# Continuation of education after marriage and its relationship with professional maternal healthcare utilization among young adult women in Bangladesh

**Sihab Howlader, Md. Aminur Rahman, Md. Mosfequr Rahman** [ID] *

Department of Population Science and Human Resource Development, University of Rajshahi, Rajshahi, Bangladesh

* mosfeque@ru.ac.bd

**Data Availability Statement:** Data are available in a public, open access repository. Data are available on the website (https://dhsprogram.com/data/Access-Instructions.cfm).

## Abstract

The relationship between women's education and the utilization of adequate maternal healthcare services has been well documented. However, the literature on how the continuation of women's post-marital education affects the utilization of maternal healthcare services is limited. Therefore, this study investigates such relationships. This study aims to examine the association between the continuation of education after marriage and the utilization of antenatal care (ANC) ($\geq$ 4 ANC, a four-contact model; and $\geq$ 8 ANC, an eight-contact model) and delivery assistance received from skilled professionals among currently married young adult women in Bangladesh. This was a cross-sectional study of 1,731 young adult women aged 15–29 years from the Bangladesh Demographic and Health Survey, 2017–18. We adopted a multivariable logistic regression analysis to examine the relationships of interest. Results show that 60.9% of women received four or more professional ANCs, 15.5% received eight or more professional ANCs, and 69.9% received professional delivery care. Compared to young adult women who did not continue their education after marriage, women who continued were more likely to utilize $\geq$4 professional ANC (adjusted odds ratio [AOR] = 1.47; 95% confidence interval [CI] = 1.11–1.94), $\geq$8 professional ANC (AOR = 1.22; 95% CI = 1.01–1.74), and professional delivery care services (AOR = 1.78; 95% CI = 1.29–2.44). In addition, age at marriage, exposure to television, and the wealth index were also found to be associated with the utilization of professional maternal healthcare services. This finding implies that implementing policies and programs that encourage girls to continue their education after marriage could potentially increase the utilization of professional ANC and delivery care services in Bangladesh.

## Introduction

The improvement of maternal health is one of the key objectives for governments globally. In spite of the notable decline in the global maternal mortality ratio (MMR) during the past two

**Funding:** The author(s) received no specific funding for this work.

**Competing interests:** None declared.

decades, with a decrease from 339 per 100,000 live births in 2000 to 223 per 100,000 live births in 2020, representing a reduction of 34.3%, it is nevertheless significant to recognize that approximately 800 women continue to die every day as a result of complications related to pregnancy and childbirth [1]. This suggests that additional attention is required to achieve the global target of reducing MMR as outlined in the United Nations Sustainable Development Goal (SDG) 3.1, which aims for an MMR of less than 70 deaths per 100,000 live births by the year 2030. The information available demonstrates that the majority of maternal deaths resulting from complications related to pregnancy and childbirth in low- and middle-income countries are from preventable causes, notably insufficient and substandard maternal healthcare [2]. Various interventions have been implemented worldwide to decrease maternal mortality rates. These interventions include strategies such as increasing the number of antenatal care (ANC) visits and delivery assistance by medically trained professionals, as well as expanding access to in-facility births [3].

Antenatal care (ANC) and delivery care services are essential measures aimed at reducing maternal mortality and are integral components of promoting safe motherhood. The clinical justification for ANC and delivery care provided by healthcare professionals is unassailable. The presence of skilled birth attendants, including midwives, doctors, and nurses who have received comprehensive training in managing uncomplicated pregnancies, childbirth, and the immediate postnatal period, as well as identifying, managing, or referring complications in both the mother and newborn [4], is crucial for ensuring optimal survival and safety outcomes for pregnant women and their newborn [5, 6]. In 2002, the World Health Organization (WHO) suggested a four-contact antenatal care model, known as focused or basic antenatal care, for uncomplicated pregnancies [7]. However, in 2016, WHO released antenatal care guidelines that proposed a comprehensive package of care to be provided through eight scheduled antenatal contacts at defined gestational weeks designed for the routine care of healthy pregnant women and adolescent girls [8]. This model offers sufficient information for women to adequately prepare for childbirth or any potential complications. It also provides life-saving information for both mother and child and creates an opportunity for healthcare personnel to have a better understanding of the pregnant woman's conditions and detect potential complications. However, the current protocol in Bangladesh adheres to the four-contact ANC model, albeit with minor variations in timing [9].

According to the Bangladesh Demographic and Health Survey 2017–18, 84% of pregnant women received at least one ANC, 55% received four or more ANCs, and 74% of births were attended by skilled healthcare personnel [10]. Despite significant progress in reducing maternal and child mortality rates, the government of Bangladesh remains highly committed to achieving the Sustainable Development Goals pertaining to the reduction of maternal and child deaths. Hence, researchers are consistently investigating the factors that contribute to increasing access to ANC and delivery care services provided by skilled personnel, with the ultimate goal of reducing maternal mortality.

Previous studies across the globe, including studies conducted in Bangladesh, have already identified several sociodemographic and psychosocial factors that have potential effects on the utilization of four or eight-contacts of ANC and skilled attendance during delivery. These factors include maternal age, parity, wealth index, place of residence, media exposure, women's empowerment and, maternal pregnancy intention status [11–20]. Maternal education has been consistently recognized as a significant factor in several studies, demonstrating positive associations with reproductive health outcomes such as utilization of antenatal care and delivery care services [12, 18–24]. Women who possess higher levels of education tend to exhibit an enhanced understanding of reproductive health, display a greater inclination towards seeking antenatal care, and experience improved pregnancy outcomes [21]. Nevertheless, women

frequently discontinue their education post-marriage [25], a phenomenon particularly pronounced in the South Asian context [26]. Prior studies have demonstrated that the cessation of education restricts economic empowerment, particularly regarding the capacity to generate independent income, which is sometimes exacerbated by the circumstances of early marriage [27, 28]. The ramifications of ceasing education encompass diminished access to sexual and reproductive health information and services, social isolation from peers and mentors, and a decline in social mobility, highlighting economic fragility [25, 28, 29]. In many societies, such as Bangladesh, women's access to education is often constrained after marriage due to prevailing traditional norms and cultural expectations [30, 31]. The continuation of education after marriage may have significant ramifications on multiple facets of women's lives, including their ability to obtain essential healthcare services like antenatal care [32].

Child marriage is a prevailing issue in the South Asian region, where a significant proportion of women, almost 30% of women aged 20–24, get married before reaching the age of 18 [33]. This occurrence frequently takes place at a stage in their lives when their education remains unfinished. The impact of marriage on a girl's education is frequently recognized as a prominent factor leading to its early termination [25]. This phenomenon is particularly evident in the South Asian context, where girls are typically withdrawn from educational institutions upon the arrangement of their marriages [26]. In the context of Bangladesh, a significant proportion of women entered into marriage prior to reaching the age of 18, during the period in which they were still actively pursuing their education at the high school or college level. According to recent data from a nationally representative study, it was found that 59% of women in the age group 20 to 24 were getting married before turning 18 [10]. Girls who are compelled to enter into early marriage often face pressure to discontinue their studies, resulting in a decrease in their educational and economic prospects [34]. While numerous studies have examined the relationship between maternal education and the utilization of ANC and delivery care services, it is worth noting that no existing study has specifically investigated the effects of women's post-marriage continuation of education on their utilization of recommended ANC and delivery assistance from medically trained professionals. Therefore, using nationally representative data, this study seeks to examine the association between the continuation of education after marriage and the utilization of ANC ($\geq$ 4 ANC, a four-contact model; and $\geq$ 8 ANC, an eight-contact model) and delivery assistance received from skilled professionals among currently married young adult women aged 15–29 years in Bangladesh. Comprehending these associations might be beneficial for policymakers in formulating effective public health programs and interventions aimed at increasing the utilization of professional ANC and delivery care services while simultaneously reducing maternal and neonatal deaths.

## Methods

### Data extraction

We utilized data from the 2017–2018 Bangladesh Demographic Health Survey (BDHS), which is a nationally representative cross-sectional survey encompassing the entire population residing in non-institutional dwelling units across the country. The BDHS survey employed a stratified, two-stage household sampling technique. In the first step, a total of 675 enumeration areas (EAs) were selected, comprising 250 EAs in urban areas and 425 EAs in rural areas. The selection process was conducted using a probability proportional to the size of the EA. In the second round of sampling, an average of 30 households per EA were independently collected from both urban and rural areas, as well as from each of the eight divisions. This was done with the aim of obtaining statistically robust estimates of key demographic and health

indicators that are representative of the entire country. The survey utilized a two-stage strati-
fied sampling approach to select households. Out of the 20,376 ever-married women aged 15–
49 who were eligible for participation in the study, a total of 20,127 women were successfully
interviewed, yielding a remarkable response rate of 99%. There was no substantial variation in
response rates observed between urban and rural residents. Each of these interviewed women
furnished data pertaining to their personal information, their children, and their households.
The data is readily available for the public and can be accessed from the MEASURE DHS data-
base at http://dhsprogram.com/data/available-datasets.cfm. However, we obtained permission
from the DHS archive to utilize the BDHS 2017–2018 data for our study. Further information
regarding the techniques for data collection and management can be obtained elsewhere [10].
The present analysis focused specifically on young adult women aged 15–29 who were cur-
rently married at the time of the survey. The rationale behind selecting young adult women
was to minimize recall bias and underreporting of educational continuation after marriage.
Therefore, the sample utilized for the present study comprised 1,731 young adult women, aged
15–29 years, who had at least one live birth in the three years preceding the survey.

## Outcome variables

We chose two indicators as our primary outcomes of interest: the utilization of professional
antenatal care and the utilization of professional delivery care. Professional maternal health-
care utilization was defined in this study based on the definitions provided in the 2017–2018
BDHS [10]. It encompassed the utilization of maternal healthcare services by healthcare pro-
fessionals who possess the necessary qualifications, such as doctors, nurses, midwives, para-
medics, family welfare visitors, community skilled birth attendants, medical assistants, or
sub-assistant community medical officers. The measurement of professional antenatal utiliza-
tion was based on the frequency of visits to healthcare professionals for antenatal care. The
participants were categorized into two groups based on WHO recommendations: (i) women
who had at least four professional ANC visits ($\geq$ 4 professional ANC), and (ii) women who
received at least eight professional ANC visits ($\geq$ 8 professional ANC). The other measure,
professional delivery care, pertained to childbirth that was facilitated by a qualified doctor,
nurse, midwife, paramedic, family welfare visitor, or community-skilled birth attendant [10].

## Exposure variable

The continuation of post-marriage education was the exposure variable of interest in this
study. The measurement of post-marriage continuation of education in the 2017–2018 BDHS
involved questioning women about their educational pursuits after being married. The ques-
tion posed to women was: "Did you continue your studies after marriage?" The available
response options were: no; yes, for less than a year; yes, for 1–2 years; yes, for 3–4 years; and
yes, for 5+ years. In the present study, this variable was categorized into two distinct groups:
"no," indicating women who did not pursue further education after getting married, and
"yes," indicating those who continued their education after marriage.

## Covariates

This investigation incorporated several theoretically relevant variables that have been previ-
ously identified as being associated with the utilization of ANC and delivery care [11–24].
These include: age of the respondent (classified into three groups: 15–19 years, 20–24 years,
and 25–29 years), age at first marriage (categorized as <18 years and $\geq$18 years), respondent's
working status (yes or no), age difference between spouses (<5 years, 5–10 years, or 11 years
and above), pregnancy intention (planned and mistimed/unwanted), and wealth index

(poorest, poorer, middle, richer, or richest). The determination of the wealth index involved categorizing respondents according to their household scores, which incorporate factors such as durable consumer items, housing quality, and water and sanitation facilities [35]. The measurement of decision-making power within households was conducted by assessing replies to specific questions regarding the individuals responsible for making decisions in the respondent's household pertaining to: (1) obtaining health care; (2) large household purchases; and (3) visits to family or relatives. The available response alternatives included: (a) respondent alone; (b) respondent and husband/partner; (c) respondent and other person; (d) husband/partner alone; (e) someone else; f) other. A value of 1 was assigned to each question if the response corresponds to options (a), (b), or (c), while a value of 0 was allocated if the response corresponds to options (d), (e), or (f). The values were then added, yielding a composite score ranging from 0 to 3 (Cronbach α is 0.80). Media access was operationalized in this study by employing three distinct mass media variables: frequency of watching television, frequency of listening to radio, and frequency of reading newspapers. The respondents were asked how frequently they read newspapers or magazines, listened to radio, or watched television, with the alternatives being not at all, less than once a week, or at least once a week. The analysis also encompassed the variables of place of residence (urban or rural) and region (Barisal, Chittagong, Dhaka, Khulna, Mymensingh, Rajshahi, Rangpur, or Sylhet).

### Ethics approval

The BDHS 2017–18 was approved by International Institutional Review Boards at ICF (ICF IRB FWA00000845) and the Bangladesh Medical Research Council (BMRC) (BMRC/NREC/2016-2019/324). The BDHS conformed to international ethical standards of confidentiality, anonymity and informed consent. This study did not require further ethics approval because it used retrospective publicly available data.

### Statistical analyses

The sample sociodemographic characteristics were described by employing weighted percentages. We utilized χ2 tests to investigate the associations between the continuation of education after marriage and the utilization of professional ANC and delivery care services, as well as other individual and household characteristics. For all analyses conducted, the significance level was established at $p < 0.05$ (two-tailed). Multivariable logistic regression was used to determine the relationships between the continuation of education after marriage and the utilization of professional ANC ($\geq 4$ professional ANC and $\geq 8$ professional ANC) and professional delivery care while controlling for theoretically relevant variables. We estimated the odds ratios (ORs) to assess the strength of the associations adjusted for potential confounders and used the 95% confidence intervals (CIs) to test the statistical significance. The examination of the variance inflation factors was used to assess the presence of multicollinearity among the variables. In all cases, the values were found to be below 2.0, suggesting a low level of multicollinearity. The statistical analyses were conducted using Stata version 13.0/MP (Stata Corp., LP, College Station, Texas, USA), taking into account sample weighting related to the complex design of the DHS.

### Results

Table 1 shows the sociodemographic and household characteristics of the sample. The average age of the respondents was 22.30 (standard deviation [SD]: 3.54), and the majority of the young adult women were married before reaching their 18th birthday (73.2%). Nearly one-third of the respondents reported that they were currently working (32.3%), and 15.8%

**Table 1. Socio-demographic and other characteristics of currently married young-adult women aged 15–29 years (n = 1731), Bangladesh Demographic and Health Survey, 2017–18.**

| Variables | Categories | Number (n) | Percent (%)[a] | 95% Confidence interval |
|---|---|---|---|---|
| **Respondent age** (Mean 22.30, (SD:3.54)) Range: 15–29 | 15–19 | 455 | 27.1 | 24.8–29.6 |
| | 20–24 | 764 | 44.5 | 41.8–47.3 |
| | 25–29 | 512 | 28.3 | 26.0–30.8 |
| **Age at first marriage** | <18 years | 1248 | 73.2 | 70.6–75.6 |
| | ≥18 years | 483 | 26.8 | 24.4–29.4 |
| **Respondent working status** | Yes | 569 | 32.3 | 29.4–35.2 |
| | No | 1162 | 67.7 | 64.8–70.6 |
| **Age difference between husband and wife** | <5 years | 318 | 17.9 | 16.0–20.0 |
| | 5–10 years | 935 | 53.2 | 50.6–55.8 |
| | ≥11 years | 478 | 28.9 | 26.4–31.5 |
| **Pregnancy intention** | Planned | 1439 | 84.2 | 82.3–86.0 |
| | Mistimed or unwanted | 292 | 15.8 | 14.0–17.7 |
| **Currently residing with husband** | Yes | 1306 | 73.3 | 70.5–76.0 |
| | No | 425 | 26.7 | 24.0–29.5 |
| **Continuation of education after marriage** | No | 1190 | 71.0 | 68.2–73.6 |
| | Yes | 541 | 29.0 | 26.4–31.8 |
| **Household decision-making power index** | 0 of 3 items | 313 | 18.6 | 16.6–20.8 |
| | 1 of 3 items | 300 | 18.2 | 16.2–20.3 |
| | 2 of 3 items | 261 | 14.9 | 13.1–17.0 |
| | All 3 items | 857 | 48.3 | 45.5–51.2 |
| **Frequency of reading newspaper/magazine** | Not at all | 1418 | 84.2 | 82.1–86.0 |
| | Less than once a week | 220 | 11.1 | 9.5–12.8 |
| | At least once a week | 93 | 4.7 | 3.7–6.0 |
| **Frequency of listening to radio** | Not at all | 1547 | 89.7 | 88.0–91.3 |
| | Less than once a week | 119 | 6.5 | 5.3–7.9 |
| | At least once a week | 65 | 3.8 | 2.9–5.0 |
| **Frequency of watching television** | Not at all | 493 | 27.1 | 24.5–29.9 |
| | Less than once a week | 139 | 8.2 | 6.9–9.7 |
| | At least once a week | 1099 | 64.7 | 61.7–67.5 |
| **Wealth index** | Poorest | 221 | 11.8 | 10.0–13.9 |
| | Poorer | 283 | 16.7 | 14.8–18.7 |
| | Middle | 327 | 19.9 | 17.6–22.3 |
| | Richer | 426 | 25.2 | 22.6–28.0 |
| | Richest | 474 | 26.5 | 23.6–29.5 |
| **Place of Residence** | Urban | 621 | 27.2 | 25.0–29.6 |
| | Rural | 1110 | 72.8 | 70.4–75.0 |
| **Region** | Barisal | 200 | 6.0 | 5.1–6.9 |
| | Chittagong | 283 | 20.9 | 18.6–23.3 |
| | Dhaka | 249 | 24.6 | 22.0–27.4 |
| | Khulna | 254 | 12.8 | 11.4–14.3 |
| | Mymensingh | 175 | 6.8 | 5.8–8.0 |
| | Rajshahi | 236 | 14.4 | 12.7–16.3 |
| | Rangpur | 222 | 11.4 | 10.1–13.0 |
| | Sylhet | 112 | 3.1 | 2.6–3.7 |

Note:

[a]In estimating percentages, the complex survey design and sampling weights were taken into account

reported their last birth as mistimed or unwanted. The majority of the respondents in this sample (71.0%) reported that they did not continue their education after marriage. Nearly one in every four young adult women did not watch television (24.3%), 84.2% did not read newspapers or magazines, and 89.7% did not listen to radio. The majority of the respondents were from rural areas (72.8%) (Table 1). Fig 1 presents the prevalence of professional ANC and delivery care among young adult women in Bangladesh. Fig 1 shows that 60.9% of women received four or more professional ANCs, 15.5% received eight or more professional ANCs, and 69.9% received professional delivery care. Fig 2 displays the regional differences in professional ANC and delivery care.

Table 2 presents the results of the bivariate analysis examining the differentials in the utilization of professional antenatal care (ANC) and delivery care services by young adult women's continuation of education after marriage and other sociodemographic variables. Utilization of ≥4 professional ANC (71.5% vs. 56.6%; p<0.001), ≥8 professional ANC (21.0% vs. 13.3%; p<0.001), and professional delivery care (82.7% vs. 64.7%; p<0.001) was higher among women who continued education after marriage. Women who were married before age 18 were less likely to use ≥4 professional ANC (59.2% vs. 65.6%; p = 0.049), ≥8 professional ANC (14.3% vs. 18.9%; p = 0.038), and professional delivery care (65.4% vs. 82.0%; p<0.001)

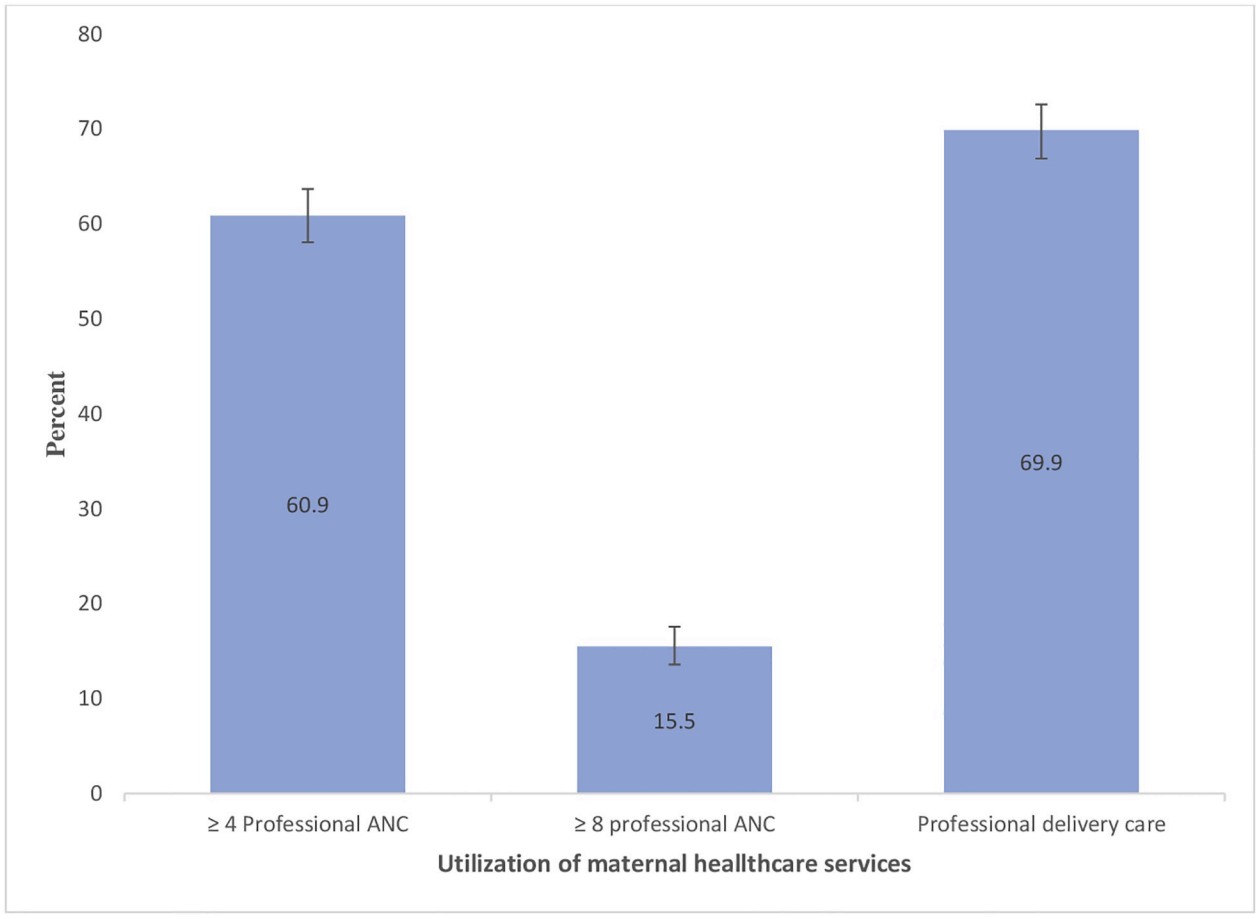

**Fig 1. Prevalence of professional maternal healthcare utilization among currently married young adult women, Bangladesh Demographic and Health Survey 2017–2018.**

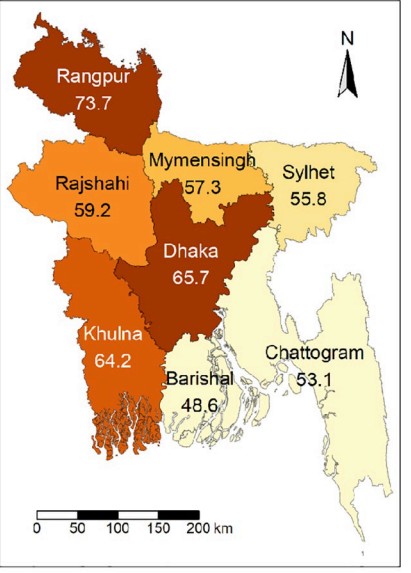
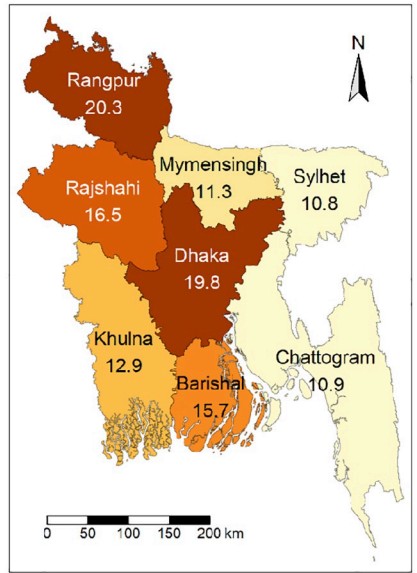

$\geq 4$ professional ANC                    $\geq 8$ professional ANC

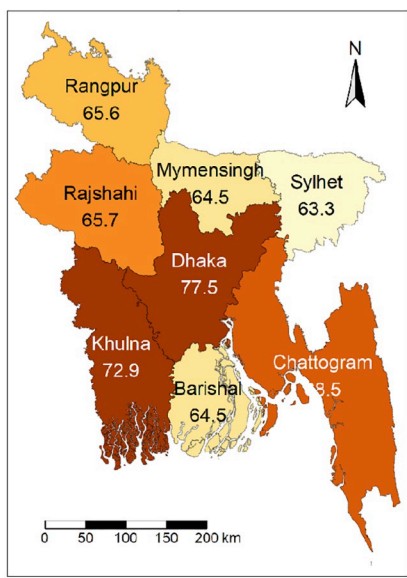

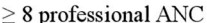

Professional delivery care

**Fig 2. Professional maternal healthcare utilization by continuation of education after marriage among currently married young adult women, Bangladesh Demographic and Health Survey 2017–2018.**

than women who were married after 18 years. Additionally, the utilization of $\geq 4$ professional ANC, $\geq 8$ professional ANC, and professional delivery services was significantly higher among individuals with access to radio, newspapers, and television at least once a week, those from the richest households, and residents of urban areas compared to their respective counterparts. Fig 3 shows the utilization of $\geq 4$ professional ANC, $\geq 8$ professional ANC, and professional delivery care by continuation of education.

**Table 2. Descriptive statistics of currently married young adult aged 15 to 29 years, by different sociodemographic and community-level variables, Bangladesh Demographic and Health Survey, 2017–18.**

| Characteristics | ≥4ANC[1] | | ≥8ANC[2] | | Professional delivery care | |
|---|---|---|---|---|---|---|
| **Continuation of education** | Yes n (%)[a] | No n (%)[a] | Yes n (%)[a] | No n (%)[a] | Yes n (%)[a] | No n (%)[a] |
| No | 677(56.6) | 513(43.4) | 164(13.3) | 1026(86.7) | 758(64.7) | 432(35.3) |
| Yes | 387(71.5) | 154(28.5) | 117(21) | 424(79.0) | 453(82.7) | 88(17.3) |
| **Exact p-value** | <0.001 | | <0.001 | | <0.001 | |
| **Respondent age** | | | | | | |
| 15–19 | 271(59.8) | 184(40.2) | 57(11.8) | 398(88.2) | 300(66.7) | 155(33.3) |
| 20–24 | 452(58.3) | 312(41.8) | 129(15.2) | 635(84.8) | 556(72.6) | 208(27.4) |
| 25–29 | 341(66.3) | 171(33.8) | 95(19.5) | 417(80.5) | 355(68.8) | 157(31.2) |
| **Exact p-value** | 0.052 | | 0.008 | | 0.124 | |
| **Age at first marriage** | | | | | | |
| <18 years | 738(59.2) | 510(40.8) | 187(14.3) | 1061(85.7) | 813(65.4) | 435(34.6) |
| ≥18 years | 326(65.6) | 157(34.4) | 94(18.9) | 389(81.1) | 398(82.0) | 85(18.0) |
| **Exact p-value** | 0.049 | | 0.038 | | <0.001 | |
| **Respondent working status** | | | | | | |
| Yes | 358(60.5) | 211(39.5) | 82(13.7) | 487(86.3) | 354(61.5) | 215(38.5) |
| No | 706(61.1) | 456(38.9) | 199(16.4) | 963(83.6) | 857(73.9) | 305(26.1) |
| **Exact p-value** | 0.826 | | 0.206 | | <0.001 | |
| **Age difference between husband and wife** | | | | | | |
| <5 years | 193(60.4) | 125(39.6) | 58(17.6) | 260(82.4) | 212(66.6) | 106(33.4) |
| 5–10 years | 558(58.9) | 377(41.1) | 151(15.8) | 784(84.2) | 646(68.9) | 289(31.1) |
| ≥11 years | 313(65.0) | 165(35.0) | 72(13.7) | 406(86.3) | 353(73.7) | 125(26.3) |
| **Exact p-value** | 0.107 | | 0.387 | | 0.116 | |
| **Pregnancy Intention** | | | | | | |
| Planned | 885(61.0) | 554(39.0) | 242(16.1) | 1197(83.9) | 1016(70.4) | 423(29.6) |
| Mistimed or unwanted | 179(60.3) | 113(39.7) | 39(12.3) | 253(87.7) | 195(67.2) | 97(32.8) |
| **Exact p-value** | 0.837 | | 0.132 | | 0.351 | |
| **Currently residing with Husband** | | | | | | |
| Yes | 822(62.6) | 484(37.4) | 218(16.2) | 1088(83.9) | 910(69.4) | 396(30.7) |
| No | 242(56.3) | 183(43.7) | 63(13.8) | 362(86.2) | 301(71.4) | 124(28.6) |
| **Exact p-value** | 0.033 | | 0.292 | | 0.473 | |
| **Household decision-making power index** | | | | | | |
| 0 of 3 items | 181(56.2) | 132(43.9) | 49(14.0) | 264(86.0) | 199(64.1) | 114(35.9) |
| 1 of 3 items | 176(58.9) | 124(41.1) | 41(14.1) | 259(85.9) | 215(70.6) | 85(29.4) |
| 2 of 3 items | 170(64.6) | 91(35.4) | 41(15.7) | 220(84.3) | 190(72.4) | 71(27.6) |
| All 3 items | 537(62.4) | 320(37.6) | 150(16.6) | 707(83.4) | 607(71.1) | 250(28.9) |
| **Exact p-value** | 0.213 | | 0.725 | | 0.135 | |
| **Frequency of reading newspaper/magazine** | | | | | | |
| Not at all | 833(58.7) | 585(41.3) | 209(14.0) | 1209(86.0) | 949(68.0) | 469(32.0) |
| Less than once a week | 154(69.7) | 66(30.3) | 41(17.7) | 179(82.3) | 181(79.2) | 39(20.8) |
| At least once a week | 77(80.8) | 16(19.2) | 31(38.0) | 62(62.0) | 81(81.8) | 12(18.3) |
| **Exact p-value** | <0.001 | | <0.001 | | 0.001 | |
| **Frequency of listening to radio** | | | | | | |
| Not at all | 930(59.7) | 617(40.3) | 232(14.3) | 1315(85.7) | 1064(69.1) | 483(30.9) |
| Less than once a week | 82(68.8) | 37(31.2) | 26(21.1) | 93(78.9) | 94(77.0) | 25(23.0) |
| At least once a week | 52(75.5) | 13(24.5) | 23(34.9) | 42(65.1) | 53(77.4) | 12(22.6) |
| **Exact p-value** | 0.021 | | <0.001 | | 0.127 | |

(*Continued*)

**Table 2.** (Continued)

| Characteristics | ≥4ANC[1] | | ≥8ANC[2] | | Professional delivery care | |
|---|---|---|---|---|---|---|
| Continuation of education | Yes n (%)[a] | No n (%)[a] | Yes n (%)[a] | No n (%)[a] | Yes n (%)[a] | No n (%)[a] |
| **Frequency of watching television** | | | | | | |
| Not at all | 242 (50.5) | 251 (49.5) | 47 (9.8) | 446 (90.2) | 265 (53.8) | 228 (46.2) |
| Less than once a week | 79 (56.0) | 60 (44.0) | 16 (9.6) | 123 (90.4) | 92 (64.2) | 47 (35.8) |
| At least once a week | 743 (65.9) | 356 (34.1) | 218 (18.7) | 881 (81.3) | 854 (77.3) | 245 (22.7) |
| **Exact p-value** | <0.001 | | <0.001 | | <0.001 | |
| **Wealth index** | | | | | | |
| Poorest | 105(48.6) | 116(51.4) | 16(7.0) | 205(93.0) | 109(50.5) | 112(49.5) |
| Poorer | 147(51.8) | 136(48.2) | 37(13.0) | 246(87.0) | 152(54.3) | 131(45.7) |
| Middle | 195(59.0) | 132(41.1) | 55(15.2) | 272(84.8) | 225(68.4) | 102(31.6) |
| Richer | 267(60.8) | 159(39.2) | 63(13.6) | 363(86.4) | 309(72.8) | 117(27.2) |
| Richest | 350(73.8) | 124(26.2) | 110(22.9) | 364(77.1) | 416(86.6) | 58(13.4) |
| **Exact p-value** | <0.001 | | <0.001 | | <0.001 | |
| **Place of Residence** | | | | | | |
| Urban | 426(67.1) | 195(32.9) | 137(20.9) | 484(79.1) | 488(80.1) | 133(19.9) |
| Rural | 638(58.6) | 472(41.4) | 144(13.5) | 966(86.5) | 723(66.1) | 387(33.9) |
| **Exact p-value** | 0.005 | | <0.001 | | <0.001 | |
| **Region** | | | | | | |
| Barisal | 100(48.6) | 100(51.4) | 37(15.7) | 163(84.4) | 134(64.5) | 66(35.5) |
| Chittagong | 151(53.1) | 132(46.9) | 31(10.9) | 252(89.1) | 195(68.5) | 88(31.5) |
| Dhaka | 165(65.7) | 84(34.3) | 52(19.8) | 197(80.2) | 196(77.5) | 53(22.5) |
| Khulna | 168(64.2) | 86(35.8) | 41(12.9) | 213(87.1) | 187(72.9) | 67(27.1) |
| Mymensingh | 104(57.3) | 71(42.7) | 21(11.3) | 154(88.7) | 114(64.5) | 61(35.5) |
| Rajshahi | 145(59.2) | 91(40.8) | 41(16.5) | 195(83.5) | 159(65.7) | 77(34.3) |
| Rangpur | 166(73.7) | 56(26.3) | 46(20.3) | 176(79.7) | 149(65.6) | 73(34.4) |
| Sylhet | 65(55.8) | 47(44.2) | 12(10.8) | 100(89.2) | 77(63.3) | 35(36.7) |
| **Exact p-value** | <0.001 | | 0.019 | | 0.050 | |
| **Total** | 1064 (60.9) | 667 (39.0) | 281 (15.5) | 1450 (84.5) | 1211 (69.9) | 520 (30.1) |

Note:

[1]at least four professional antenatal care;

[2]at least eight professional antenatal care

Table 3 presents the findings of multivariable logistic regression analyses that explore the relationships between women's continuation of education after marriage and the utilization of professional antenatal care (ANC) and delivery care services. The results indicate that compared to young adult women who did not continue education after marriage, women who continued education after marriage were 1.47 times (adjusted odds ratio [AOR] = 1.47; 95% confidence interval [CI] = 1.11–1.94), 1.22 times (AOR = 1.22; 95% CI = 1.01–1.74), and 1.78 times (AOR = 1.78; 95% CI = 1.29–2.44) more likely to utilize ≥4 professional ANC, ≥8 professional ANC, and professional delivery care services, respectively. In addition, utilization of professional delivery care services was 78% higher among young adult women who were married at or after age 18 than among young adult women who married before age 18. Women who watched television at least once a week were more likely to utilize ≥8 professional ANC (AOR = 1.52; 95% CI = 1.04–4.95) and professional delivery care (AOR = 1.76; 95% CI = 1.28–2.40) than their counterparts who had not watched television at all. Moreover, compared to

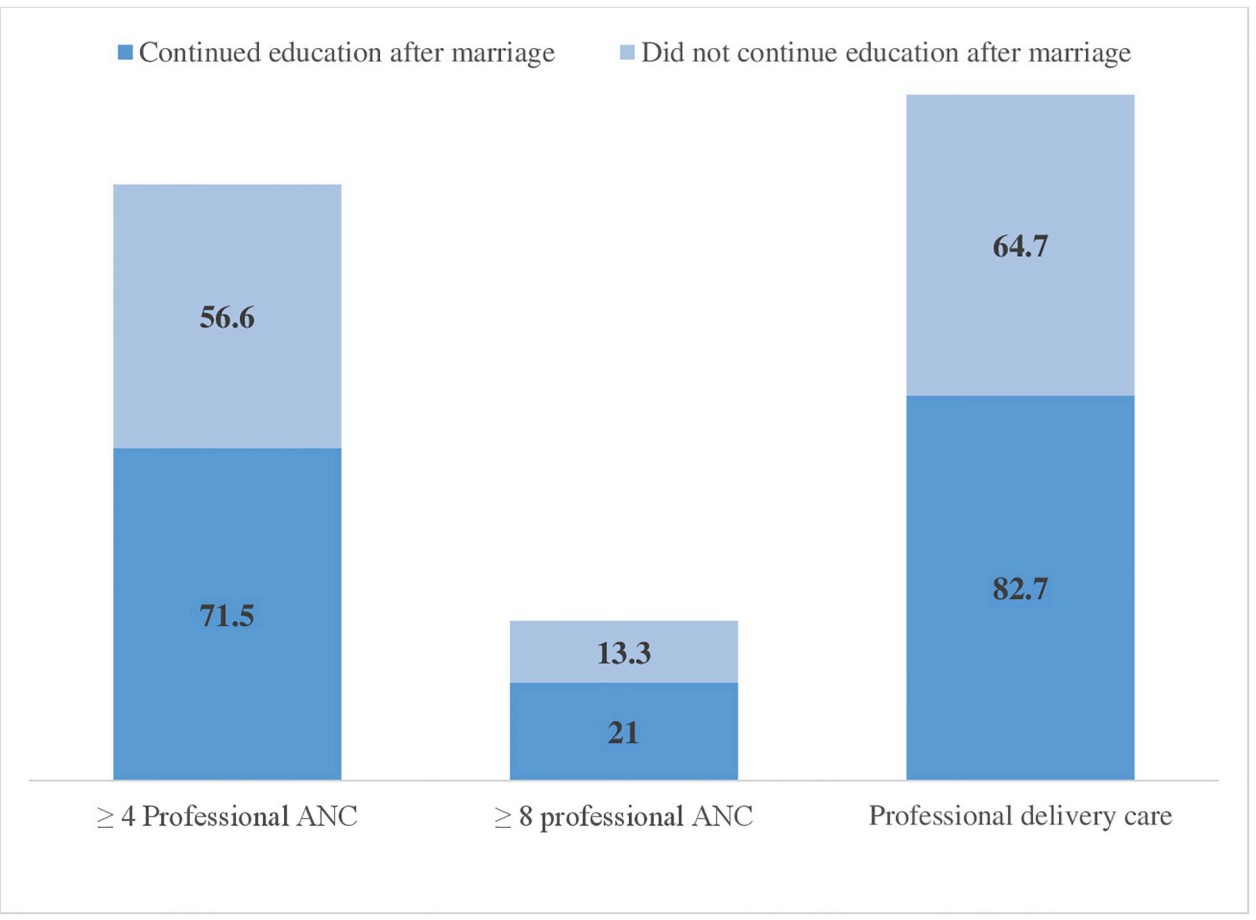

**Fig 3. Regional distribution of professional maternal healthcare utilization among currently married young adult women, Bangladesh Demographic and Health Survey 2017–2018.**

women from the poorest household, women from the richest household were more likely to utilize ≥4 professional ANC (AOR = 3.04; 95% CI = 1.85–4.97), ≥8 professional ANC (AOR = 2.91; 95% CI = 1.41–5.98), and professional delivery care services (AOR = 3.48; 95% CI = 2.13–5.67).

## Discussion

This study aims to assess the relationships between the continuation of education after marriage and the utilization of professional maternal healthcare services, specifically ≥4 professional ANC, ≥8 professional ANC, and professional delivery care services. This study found that 60.9%, 15.5%, and 69.9% of currently married young adult women aged 15–29 who had at least one live birth in the three years preceding the survey received ≥4 professional ANC services, ≥8 professional ANC services, and professional delivery care services, respectively. Furthermore, this study demonstrates that a mere 29.0% of young adult women continued their education after marriage. This study also suggests that young adults' continuation of education after marriage is associated with professional maternal healthcare utilization (≥4 professional ANC, ≥8 professional ANC, and professional delivery care services) with higher odds.

**Table 3. Results from multivariable logistic regression analysis examining the factors associated with continuation of education among currently married young adult aged 15–29 years, Bangladesh Demographic and Health Survey, 2017–18.**

| Characteristics | ≥4ANC[1] | ≥8ANC[2] | Delivery care |
|---|---|---|---|
| **Continuation of education** | OR (95% CI) | OR (95% CI) | OR (95% CI) |
| No | 1.0 | 1 | 1.0 |
| Yes | 1.47 (1.11–1.94) | 1.22 (1.01–1.74) | 1.78 (1.29–2.44) |
| **Respondent age** | | | |
| 15–19 | 1.0 | 1.0 | 1.0 |
| 20–24 | 0.79 (0.58–1.08) | 1.20 (0.81–1.77) | 0.96 (0.68–1.34) |
| 25–29 | 1.13 (0.79–1.58) | 1.57 (1.02–2.42) | 0.78 (0.55–1.10) |
| **Age at first marriage** | | | |
| <18 years | 1.0 | 1.0 | 1.0 |
| ≥18 years | 1.02 (0.75–1.38) | 0.90 (0.61–1.33) | 1.78 (1.26–2.52) |
| **Respondent working status** | | | |
| No | 1.0 | 1.0 | 1.0 |
| Yes | 1.09 (0.83–1.43) | 0.89 (0.63–1.25) | 0.80 (0.60–1.07) |
| **Age difference between husband and wife** | | | |
| <5 years | 1.0 | 1.0 | 1.0 |
| 5–10 years | 0.98 (0.73–1.31) | 0.93 (0.63–1.38) | 1.07 (0.77–1.49) |
| ≥11 years | 1.34 (0.95–1.90) | 0.81 (0.52–1.28) | 1.27 (0.85–1.89) |
| **Pregnancy Intention** | | | |
| Planned | 1.0 | 1.0 | 1.0 |
| Mistimed or unwanted | 0.94 (0.69–1.27) | 0.72 (0.46–1.15) | 0.81 (0.58–1.14) |
| **Currently residing with husband** | | | |
| Yes | 1.0 | 1.0 | 1.0 |
| No | 0.83 (0.63–1.08) | 0.93 (0.65–1.32) | 1.08 (0.79–1.47) |
| **Household decision-making power index** | | | |
| 0 of 3 items | 1.0 | 1.0 | 1.0 |
| 1 of 3 items | 1.12 (0.77–1.62) | 1.01 (0.58–1.75) | 1.38 (0.94–2.02) |
| 2 of 3 items | 1.27 (0.85–1.89) | 0.94 (0.54–1.60) | 1.44 (0.95–2.21) |
| All 3 items | 1.19 (0.86–1.64) | 1.00 (0.63–1.59) | 1.39 (1.01–1.91) |
| **Frequency of reading newspaper/magazine** | | | |
| Not at all | 1.0 | 1.0 | 1.0 |
| Less than once a week | 1.23 (0.81–1.87) | 0.99 (0.64–1.54) | 1.00 (0.64–1.57) |
| At least once a week | 1.85 (0.95–3.60) | 2.22 (1.24–3.97) | 0.73 (0.36–1.49) |
| **Frequency of listening to radio** | | | |
| Not at all | 1.0 | 1.0 | 1.0 |
| Less than once a week | 1.31 (0.81–2.11) | 1.49 (0.64–1.55) | 1.15 (0.67–1.95) |
| At least once a week | 1.56 (0.80–3.01) | 2.63 (1.40–4.95) | 1.09 (0.54–2.23) |
| **Frequency of watching television** | | | |
| Not at all | 1.00 | 1.00 | 1.00 |
| Less than once a week | 1.11 (0.70–1.76) | 0.87 (0.44–1.74) | 1.34 (0.85–2.08) |
| At least once a week | 1.34 (0.99–1.81) | 1.52 (1.04–4.95) | 1.76 (1.28–2.40) |
| **Wealth index** | | | |
| Poorest | 1.00 | 1.00 | 1.00 |
| Poorer | 1.18 (0.79–1.76) | 1.97 (0.98–3.92) | 1.04 (0.68–1.58) |
| Middle | 1.68 (1.13–2.49) | 2.33 (1.17–4.63) | 1.56 (1.04–2.34) |
| Richer | 1.75 (1.15–2.65) | 1.87 (0.93–3.75) | 1.72 (1.14–2.60) |
| Richest | 3.04 (1.85–4.97) | 2.91 (1.41–5.98) | 3.48 (2.13–5.67) |

*(Continued)*

**Table 3.** (Continued)

| Characteristics | ≥4ANC[1] | ≥8ANC[2] | Delivery care |
|---|---|---|---|
| **Continuation of education** | OR (95% CI) | OR (95% CI) | OR (95% CI) |
| **Place of Residence** | | | |
| Urban | 1.00 | 1.00 | 1.00 |
| Rural | 1.08 (0.82–1.42) | 0.89 (0.61–1.28) | 0.80 (0.58–1.11) |
| **Region** | | | |
| Barisal | 1.0 | 1.0 | 1.00 |
| Chittagong | 0.82 (0.51–1.32) | 0.43 (0.24–0.76) | 0.70 (0.42–1.16) |
| Dhaka | 1.35 (0.83–2.19) | 0.79 (0.44–1.42) | 0.93 (0.53–1.64) |
| Khulna | 1.59 (0.99–2.55) | 0.61 (0.34–1.10) | 1.24 (0.73–2.09) |
| Mymensingh | 1.29 (0.80–2.10) | 0.56 (0.28–1.15) | 0.88 (0.52–1.47) |
| Rajshahi | 1.32 (0.82–2.12) | 0.86 (0.47–1.56) | 0.93 (0.56–1.55) |
| Rangpur | 3.38 (2.00–5.72) | 1.44 (0.80–2.58) | 1.25 (0.71–2.20) |
| Sylhet | 1.01 (0.57–1.80) | 0.44 (0.16–1.22) | 0.61 (0.34–1.11) |

*Note*: OR, Odds ratio; CI, Confidence interval

[1]at least four or more antenatal cares,

[2]at least eight or more antenatal, CI, Confidence interval

This study found that the likelihood of utilizing professional maternal health care services is higher among women who continued their education after marriage. Although we did not find any study that directly assessed the relationship between the continuation of education after marriage and maternal healthcare utilization, a number of previous studies across the world have already documented the relationship between maternal education and antenatal care service utilization [22, 32, 36, 37]. Our findings align with the findings of these studies. The observed associations may be attributed to the notion that the continuation of education after marriage augments women's ability to participate in making decisions in the family as well as to make informed choices pertaining to their own health and well-being. Education is widely recognized as a primary and crucial aspect of empowering women [38]. Women who continued education after marriage may have an increased likelihood of becoming more well-informed and aware, thereby gaining the ability to make better decisions regarding their own health and the health of their children. Women who pursue education after getting married may possess a greater understanding of the functioning and accessibility of the healthcare system. This knowledge equips them to effectively engage and communicate with healthcare providers, enabling them to comprehend the advantages of utilizing maternal healthcare services provided by professionals. Consequently, it is expected that these women will utilize such services more efficiently [39, 40].

In addition to the continuation of education after marriage, several other factors, including age at marriage, exposure to television, and wealth index, were also found to be associated with the utilization of professional antenatal and delivery care services. In accordance with previous studies [12, 41], the present study observes a notable association between age at marriage and utilization of professional delivery care. Young adult women who married at or after the age of 18 were more likely to utilize professional delivery care services than those who married before 18. One plausible explanation for the higher utilization of maternal healthcare services among young adult women who married at age ≥18 could be attributed to their enhanced agency in decision-making regarding healthcare utilization; these decisions are generally controlled by husbands and in-laws in the South Asian context [42, 43]. Limited decision-making ability in

the household was identified in previous studies as a major barrier for the utilization of maternal health care services among young married women [39, 44, 45]. Furthermore, it is worth noting that women who enter into marriage before reaching the age of 18 may encounter many challenges, such as limited access to health information, poverty, and limited health facilities, which could lead to a lower utilization of professional maternal health care services.

Corroborating prior research [46, 47], our findings indicate that exposure to mass media, particularly watching television, is positively associated with the utilization of professional antenatal and delivery care services. The mass media plays a key role in the transmission of important health information. Hence, it can be inferred that young adult women who are exposed to various forms of media have convenient access to information that enables them to make well-informed choices regarding their own health. Consequently, this heightened awareness is likely to enhance their inclination to seek maternal health care services from qualified healthcare professionals [46]. The consumption of mass media has the potential to foster favorable attitudes towards the utilization of professional maternal health care services, as individuals are influenced by the information they have been exposed to through different forms of media [48]. In alignment with prior studies [12, 49, 50], the present study similarly reported that the likelihood of utilizing WHO-recommended professional antenatal care and professional delivery care services increases with increasing economic status as measured by the wealth index. Plausible explanations could be that young adult women with better economic status can have the ability to cover expenses related to healthcare services, including transportation, medications, and any associated costs, and they may have greater access to information regarding the benefits of obtaining the recommended number of professional ANC and delivery care services [50].

## Strengths and limitations

The main strength of this study is the use of a nationally representative sample including both rural and urban areas with a large sample size that permits precise estimates. However, the interpretation of the findings should take into account a number of limitations. Due to retrospective reporting of the continuation of education after marriage, there may be some recall bias in the data. Biases may also arise as a consequence of unanticipated educational pursuits upon marriage. Additionally, the data on utilizing ANC and delivery care services was collected based on self-reports from mothers within 3 years preceding the survey and was not confirmed by medical records, which could be a potential source of recall and misclassification bias. The sample of this study includes only young adult women aged 15–29 years, so the findings should not be generalized to women of all ages. The utilization of secondary data imposes constraints on the analysis, as it fails to incorporate various potential factors such as cost of care, availability and accessibility of health facilities, equity in health service delivery, or knowledge and attitudes towards modern healthcare services that may influence the healthcare seeking behaviors of young adult women. Finally, because of the cross-sectional design of the study, the analysis can only provide evidence of statistical association, and cause-and-effect relationships cannot be inferred.

## Implications of the study

The ramifications of the study's findings hold considerable policy significance and have wide-ranging consequences for the utilization of maternal healthcare in Bangladesh. We propose several priorities. First, the promotion of women's education should be a priority for policy-makers, since it has been found that the continuation of education after marriage has a favorable association with the utilization of recommended professional antenatal care and

delivery care services. Therefore, interventions aiming for efficient and effective utilization of maternal healthcare services should prioritize women's education, especially in the context of continuing education after marriage. In order to facilitate the pursuit of women's education after marriage, it is suggested to implement targeted stipend programs that cater specifically for disadvantageous families. Strict implementation of the legislation against child marriages could also help women stay in school for a longer duration. Second, the findings of this study suggest that exposure to mass media can significantly contribute to the utilization of recommended professional ANC and professional delivery care services; therefore, appropriate authorities should take proactive measures to disseminate maternal healthcare information pertaining to antenatal visits, healthcare at delivery, and postnatal check-ups to pregnant women more frequently through different mass media platforms such as TV, radio, and newspapers. Additionally, we recommend policymakers, healthcare providers, and health organizations implement mass media campaigns to publicize the maternal health message and motivate pregnant women to receive adequate maternal healthcare services from qualified professionals. Third, it is imperative for initiatives aimed at promoting safe motherhood to prioritize the needs of marginalized and disadvantaged women who may face barriers in accessing adequate professional antenatal and delivery care [51]. We recommend that the demand-side financing (DSF) scheme be enhanced to specifically cater to the bottom or the poorest 20% of the women. The DSF, which was initiated in 2004, is a maternal health voucher program that was designed by the Ministry of Health and Family Welfare (MOHFW) of the Bangladesh Government in collaboration with WHO and has been shown to improve access to maternal care [52].

## Conclusion

This study is a novel contribution as it is the first to document the relationship between the continuation of education after marriage and the utilization of recommended professional antenatal care and professional delivery care among young adult women in Bangladesh. We found that women's continuation of education after marriage was positively associated with several key indicators of maternal healthcare services. This finding implies that implementing policies and programs that encourage girls to continue their education after marriage could potentially increase the utilization of professional ANC and delivery care services. Bangladesh is making concerted efforts to attain the SDGs pertaining to health. This study provides a valuable addition by documenting the existence of unmet needs for antenatal care and delivery care, which serve as substantial obstacles to the attainment of the SDG 3 of reducing global maternal, neonatal, and under five mortalities. Additionally, our findings are useful to inform governments and local and international partners in other low-income countries who have been collaborating in the global effort to reduce maternal and neonatal deaths of the need to focus on the continuation of education after marriage. However, besides quantitative research, qualitative investigations should be conducted to better understand the experiences, challenges, motivations, and thought processes of women pursuing further education after marriage and their relationship with the utilization of health care services in such settings.

## Acknowledgments

The authors would like to thank MEASURE DHS Program and ICF International for giving permission to download and analyze the BDHS data.

## Author Contributions

**Conceptualization:** Sihab Howlader.

**Data curation:** Sihab Howlader.

**Formal analysis:** Sihab Howlader, Md. Aminur Rahman, Md. Mosfequr Rahman.

**Investigation:** Sihab Howlader, Md. Mosfequr Rahman.

**Methodology:** Sihab Howlader, Md. Aminur Rahman, Md. Mosfequr Rahman.

**Software:** Sihab Howlader.

**Supervision:** Md. Mosfequr Rahman.

**Visualization:** Md. Aminur Rahman.

**Writing – original draft:** Sihab Howlader, Md. Aminur Rahman, Md. Mosfequr Rahman.

**Writing – review & editing:** Sihab Howlader, Md. Mosfequr Rahman.

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
