## [Decision Letter · Decision Letter 0]

26 Aug 2024

PONE-D-24-13879Continuation of education after marriage and its relationship with professional maternal healthcare utilization among young adult women in BangladeshPLOS ONE

Dear Dr. Rahman,

Thank you for submitting your manuscript to PLOS ONE. After careful consideration, we feel that it has merit but does not fully meet PLOS ONE’s publication criteria as it currently stands. Therefore, we invite you to submit a revised version of the manuscript that addresses the points raised during the review process.

1. Revise the manuscript carefully based on the reviewer's comments and follow the journal style.

2. In Figure 1, it is suggested to show the complete percentages of ANC visits i.e., presents all categories.

3. In Figure 2, it is recommended to add the scale and direction.

4. Add strengths and limitations as a separate section.

5. Specify which goals of SDGs will be attained.

We look forward to receiving your revised manuscript.

Kind regards,

Md. Moyazzem Hossain

Academic Editor

PLOS ONE

“None declared.”

4. We note that Figure 2 in your submission contain [map/satellite] images which may be copyrighted. All PLOS content is published under the Creative Commons Attribution License (CC BY 4.0), which means that the manuscript, images, and Supporting Information files will be freely available online, and any third party is permitted to access, download, copy, distribute, and use these materials in any way, even commercially, with proper attribution. For these reasons, we cannot publish previously copyrighted maps or satellite images created using proprietary data, such as Google software (Google Maps, Street View, and Earth). For more information, see our copyright guidelines: http://journals.plos.org/plosone/s/licenses-and-copyright.

Reviewers' comments:

Reviewer's Responses to Questions

**Comments to the Author**

1. Is the manuscript technically sound, and do the data support the conclusions?

Reviewer #1: No

Reviewer #2: Yes

2. Has the statistical analysis been performed appropriately and rigorously? 

Reviewer #1: No

Reviewer #2: Yes

3. Have the authors made all data underlying the findings in their manuscript fully available?

Reviewer #1: No

Reviewer #2: Yes

4. Is the manuscript presented in an intelligible fashion and written in standard English?

Reviewer #1: No

Reviewer #2: Yes

5. Review Comments to the Author

Reviewer #1: Many thanks for inviting me to review this paper, and I am extremely sorry for the delay in submitting my revision. I read this paper with a lot of interest and care; however, I found several areas that suggest rejection of this paper for publication in PLoS ONE or any other journal.

My particular concern lies in the association that the author explored. The survey that the authors analyzed collected data on the continuation and discontinuation of education only for women who married early. I request the authors to check the survey reports and relevant questions for details. Therefore, all the samples analyzed were women who married early. Statistics related to age at first marriage in Table 1 also support this, as the reported early marriage rate is 73% compared to the national average early marriage rate of 51%, according to the same survey. However, it is unfortunate that the authors manipulated part of the data by mentioning that 26.8% did not marry early, which is impossible considering the data analysis.

In addition to this issue, considering the early marriage sample, which is mostly from the lower quintile, as representative of the total sample is not logical. In such cases, the main issue is whether these women had the knowledge and decision-making autonomy to access healthcare services rather than the continuation of education. The authors’ analysis here seems to focus on a minor issue while hiding a more important one.

Moreover, according to the government of Bangladesh, the recommended number of ANC visits is four, although WHO changed it to six to eight visits as the global recommendation. As such, it is completely misleading if the authors consider eight ANC visits as effective for Bangladesh. Additionally, the authors generated several variables, and a few of them are completely misleading, like media access categorized as "no access" and "some access." Is it possible to find people who are not connected in any way with radio, television, or newspapers that were used to generate this variable? Moreover, I am not sure how it is logical to say "some access" if respondents reported they accessed newspapers, radio, and television. Moreover, adding this group, i.e., all three access types, with the group that reported access to one or two types and classifying them together is also very problematic. It seems the authors compared a banana with an apple, which is funny instead.

The authors also generated some funny and useless maps. For instance, the first two maps regarding ANC in Figure 2 are mutually exclusive; however, they never mentioned this. Moreover, the authors used the range of their reported prevalence as the label of the maps. However, what is the explanation for this? It seems the maps were created without any statistical knowledge. Moreover, consider a typical scenario where a woman gets married at age 14 and stops her education, and the index birth analyzed occurred at age 24. Statistically, is it logical to associate the age at marriage with healthcare service access in such a case? Although I agree that if you input them into statistical software without proper statistical knowledge, you will definitely get results. This is what happened in your case. Summarizing these, the analysis done is completely useless and WRONG, although the authors did a moderate job in writing the paper with follow-up.

I believe the authors are more interested in publishing their paper rather than considering the scientific value and justification of their paper. This practice is increasingly prevalent among authors in LMICs, including Bangladesh. I request the authors to change their motivation to focus on scientific value rather than merely publication. PLEASE DO NOT WRITE SUCH MEANINGLESS PAPERS. This is never done by a genuine academician.

Reviewer #2: Dear Authors

Introduction: Expand the literature review in the introduction to provide a more comprehensive background.

Methods: the study design it is not mentioned.

Expand Future Research Directions: Suggest specific areas for future research

Thank you

6. PLOS authors have the option to publish the peer review history of their article (what does this mean?). If published, this will include your full peer review and any attached files.

Reviewer #1: No

Reviewer #2: **Yes: **Dr Mona Gamal Mohamed

---

## [Author Response · Author response to Decision Letter 0]

18 Oct 2024

18 October, 2024

Md. Moyazzem Hossain

Academic Editor

Plos One

Re: Manuscript Number PONE-D-24-13879

Dear Prof. Md. Moyazzem Hossain,

Thank you for considering and giving us the chance to submit the revision of our manuscript, Continuation of education after marriage and its relationship with professional maternal healthcare utilization among young adult women in Bangladesh, for possible publication in Plos One. We revised the manuscript based on the editor and reviewer’s feedback. Their useful and productive comments helped us improve the clarity and quality of the manuscript. Using the track change option, we have changed the text in the manuscript. It is to mention that this manuscript has not been published or presented elsewhere in part or in entirety and is not under consideration by another journal. We have read and understood the journal's policies, and we believe that neither the manuscript nor the study violates any of these. The authors have no conflicts of interest to declare. As a corresponding author, I confirm full access to all aspects of the research and writing process, and final responsibility for the paper.

We hope that the revisions are satisfactory in addressing issues raised by the reviewers and look forward to hearing your decision about this article.

Yours sincerely,

Corresponding Author

Editor’s comment

1. Revise the manuscript carefully based on the reviewer's comments and follow the journal style.

Response: Thank you for giving us the opportunity to revise the manuscript. We have carefully accommodate all the issues while revising the manuscript.

2. In Figure 1, it is suggested to show the complete percentages of ANC visits i.e., presents all categories.

Response: Thank you. We have revised it accordingly.

3. In Figure 2, it is recommended to add the scale and direction.

Response: Thank you. We have added the scale and direction in the revised version of figure 2.

4. Add strengths and limitations as a separate section.

Response: Thank you. We have separated this section in text.

5. Specify which goals of SDGs will be attained.

Response: Thank you. We have specified in the conclusion section of the text that SDG 3 will be attained. 

Response: We have revised the manuscript following PLOS ONE’s guidelines for preparing manuscript.

“None declared.”

Response: Thank you. We did it.

Response: Thank you. We have moved the ethics in the method section according to the journal’s requirement. 

4. We note that Figure 2 in your submission contain [map/satellite] images which may be copyrighted. All PLOS content is published under the Creative Commons Attribution License (CC BY 4.0), which means that the manuscript, images, and Supporting Information files will be freely available online, and any third party is permitted to access, download, copy, distribute, and use these materials in any way, even commercially, with proper attribution. For these reasons, we cannot publish previously copyrighted maps or satellite images created using proprietary data, such as Google software (Google Maps, Street View, and Earth). For more information, see our copyright guidelines: http://journals.plos.org/plosone/s/licenses-and-copyright.

Response: We prepared Figure 2 by using ggplot2 package and sf package in R.

Reviewer

Reviewer #1: Many thanks for inviting me to review this paper, and I am extremely sorry for the delay in submitting my revision. I read this paper with a lot of interest and care; however, I found several areas that suggest rejection of this paper for publication in PLoS ONE or any other journal.

Response: Thank you very much for your valuable time for reviewing our manuscript. 

My particular concern lies in the association that the author explored. The survey that the authors analyzed collected data on the continuation and discontinuation of education only for women who married early. I request the authors to check the survey reports and relevant questions for details. Therefore, all the samples analyzed were women who married early. Statistics related to age at first marriage in Table 1 also support this, as the reported early marriage rate is 73% compared to the national average early marriage rate of 51%, according to the same survey. However, it is unfortunate that the authors manipulated part of the data by mentioning that 26.8% did not marry early, which is impossible considering the data analysis.

Response: Thank you. We did review the report before beginning the analysis. However, in response to your recommendation, we rechecked the BDHS 2017-18 final report. The survey only asked women who were studying just before the marriage the question about continuation or discontinuation of education. Exact wording in the survey questionnaire was: “Were you studying or attending school just before you got married?" Response option: “Yes or No." If yes, then they were asked about how long they continued. Please see Appendix G, Page No. 361 of the final report of BHDS 2017-18.

We found no reference in the report that data on educational continuation/discontinuation were only obtained for women who got married early (before the age of 18). Rather, we observed in Table 4.10 on page 52 of the study that 80.5% of women who marry before the age of 18 quit their education, compared to 62.8% and 45% for women who marry between the ages of 18 and 20 and 21 and older, respectively. If the BDHS only collected information for women who married before the age of 18, how did they categorize such proportions? 

It's disappointing that the reviewer accused us of manipulating data. We would be pleased to offer the DO file for our analysis at any time.

In addition to this issue, considering the early marriage sample, which is mostly from the lower quintile, as representative of the total sample is not logical. In such cases, the main issue is whether these women had the knowledge and decision-making autonomy to access healthcare services rather than the continuation of education. The authors’ analysis here seems to focus on a minor issue while hiding a more important one.

Response: Thank you. First of all, data were not collected for women who got married early but rather collected based on their study or attendance at school just before marriage. The eligible women for this current study were young adults (aged 15–29 years), currently married, and who had at least one live birth in the previous three years of the survey. Our data also shows that 11.8% and 16.7% were from the poorest and poor wealth quintiles.

Ample of studies across the globe, including researchers from high-income countries, have also documented that household decision-making autonomy was found to be associated with maternal and child healthcare utilization. A very few examples are given below:

Story WT, Burgard SA. Couples' reports of household decision-making and the utilization of maternal health services in Bangladesh. Soc Sci Med. 2012 Dec;75(12):2403-11. doi: 10.1016/j.socscimed.2012.09.017. Epub 2012 Sep 26. PMID: 23068556; PMCID: PMC3523098.

Pokhrel S, Sauerborn R. Household decision-making on child health care in developing countries: the case of Nepal. Health Policy Plan. 2004 Jul;19(4):218-33. doi: 10.1093/heapol/czh027. PMID: 15208278.

Gebeyehu NA, Gelaw KA, Lake EA, Adela GA, Tegegne KD, Shewangashaw NE. Women decision-making autonomy on maternal health service and associated factors in low- and middle-income countries: Systematic review and meta-analysis. Women’s Health. 2022;18. doi:10.1177/17455057221122618

Xiaohui Hou, Ning Ma, The effect of women’s decision-making power on maternal health services uptake: evidence from Pakistan, Health Policy and Planning, Volume 28, Issue 2, March 2013, Pages 176–184, https://doi.org/10.1093/heapol/czs042

The household decision-making power/autonomy was only included as covariates based on earlier studies. Yes, there are a lot of other important issues that affect maternal healthcare utilization; however, we should not ignore any minor issues that could have potential relationships with healthcare utilization. Since education of mothers is a powerful factor in seeking healthcare, therefore, we expected that women who did not continue their education or not complete their education are less likely to utilize healthcare facilities. Therefore, we conducted this study.

Moreover, according to the government of Bangladesh, the recommended number of ANC visits is four, although WHO changed it to six to eight visits as the global recommendation. As such, it is completely misleading if the authors consider eight ANC visits as effective for Bangladesh. Additionally, the authors generated several variables, and a few of them are completely misleading, like media access categorized as "no access" and "some access." Is it possible to find people who are not connected in any way with radio, television, or newspapers that were used to generate this variable? Moreover, I am not sure how it is logical to say "some access" if respondents reported they accessed newspapers, radio, and television. Moreover, adding this group, i.e., all three access types, with the group that reported access to one or two types and classifying them together is also very problematic. It seems the authors compared a banana with an apple, which is funny instead.

Response: Thank you. Although WHO recommended eight visits for complete ANC visits in 2016, Bangladesh country recommendations continue to promote four or more ANC visits. The goal of this cross-sectional study was not to evaluate the effectiveness of eight ANC visits in Bangladesh; rather, we wanted to see how mothers' continued education was connected with greater or equivalent to eight ANC, which the government could implement in the future.

In terms of media access, I agree that it's difficult to imagine people without access to radio, TV, or newspapers. As shown on page 31 of the BDHS 2017-18 final report, over two-fifths (44%) of women are not frequently exposed to any of these kinds of media; just 3.2% listen to radio, 55% watch TV, and 2.1% read a newspaper once a week (Table 3.4, page 37). Although we may cite a number of studies in which the authors constructed media exposure, as we did here because you have objections about such categories, we present these media individually, along with their original classification, and amend our analysis.

The authors also generated some funny and useless maps. For instance, the first two maps regarding ANC in Figure 2 are mutually exclusive; however, they never mentioned this. Moreover, the authors used the range of their reported prevalence as the label of the maps. However, what is the explanation for this? It seems the maps were created without any statistical knowledge. 

Response: Thank you. In figure 2, ≥4 ANC and ≥8 ANC are not mutually exclusive. Respondents who took ≥8 ANC were also included in the category of ≥4 ANC. We removed the label of the reported prevalence, and as suggested by the academic editor, we have included scale and direction to the figure. 

Moreover, consider a typical scenario where a woman gets married at age 14 and stops her education, and the index birth analyzed occurred at age 24. Statistically, is it logical to associate the age at marriage with healthcare service access in such a case? 

Response: Thank you. In this regard, cross-sectional study designs are unhelpful. Of course, this study design has limits, but it serves as a foundation for future research and lays the groundwork for investigations into cause-and-effect linkages.

Although I agree that if you input them into statistical software without proper statistical knowledge, you will definitely get results. This is what happened in your case. Summarizing these, the analysis done is completely useless and WRONG, although the authors did a moderate job in writing the paper with follow-up.

Response: Thank you. We will be pleased to share our Stata DO file anytime for cross-checking the results . However, we appreciate that you at least you find something moderately good in the manuscript.

I believe the authors are more interested in publishing their paper rather than considering the scientific value and justification o

---

## [Decision Letter · Decision Letter 1]

6 Dec 2024

Continuation of education after marriage and its relationship with professional maternal healthcare utilization among young adult women in Bangladesh

PONE-D-24-13879R1

Dear Dr. Rahman,

We’re pleased to inform you that your manuscript has been judged scientifically suitable for publication and will be formally accepted for publication once it meets all outstanding technical requirements.

Kind regards,

Md. Moyazzem Hossain, PhD

Academic Editor

PLOS ONE

Additional Editor Comments (optional):

Reviewers' comments:

Reviewer's Responses to Questions

**Comments to the Author**

1. If the authors have adequately addressed your comments raised in a previous round of review and you feel that this manuscript is now acceptable for publication, you may indicate that here to bypass the “Comments to the Author” section, enter your conflict of interest statement in the “Confidential to Editor” section, and submit your "Accept" recommendation.

Reviewer #2: All comments have been addressed

2. Is the manuscript technically sound, and do the data support the conclusions?

Reviewer #2: (No Response)

3. Has the statistical analysis been performed appropriately and rigorously? 

Reviewer #2: Yes

4. Have the authors made all data underlying the findings in their manuscript fully available?

Reviewer #2: Yes

5. Is the manuscript presented in an intelligible fashion and written in standard English?

Reviewer #2: Yes

6. Review Comments to the Author

Reviewer #2: Dear Authors,

Greetings,

Thank you for addressing all the previous comments. However, the figures still need improvement to enhance their visibility, as they appear unclear. Kindly revise them for better clarity.

Best regards,

Dr. Mona Gamal

7. PLOS authors have the option to publish the peer review history of their article (what does this mean?). If published, this will include your full peer review and any attached files.

Reviewer #2: **Yes: **Dr. Mona Gamal Mohamed

---

## [Editor Report · Acceptance letter]

18 Dec 2024

PONE-D-24-13879R1 

PLOS ONE

Dear Dr. Rahman, 

I'm pleased to inform you that your manuscript has been deemed suitable for publication in PLOS ONE. Congratulations! Your manuscript is now being handed over to our production team.

Kind regards, 

on behalf of

Professor Md. Moyazzem Hossain 

Academic Editor

PLOS ONE